# Rare-Label-Oriented Discriminative Driven Feature Construction for Double Incomplete Multi-View Multi-Label Classification

## Abstract

The double incomplete multi-view multi-label classification(DiMvMLC) task has attracted much attention due to the prevalence of missing views and sparse labels in real-world scenarios. However, existing methods over-rely on multi-view consensus information modeling, which results in view specificity being masked and weakens the ability to recognize rare labels. To this end, this paper proposes a model based on rare-label-oriented discriminative driven feature construction method. Through a view-specific label learning strategy, shared features and private features are decoupled to enable collaborative classification modeling of multi-view characteristics guided by commonalities. Specifically, a dual feature extraction encoder is designed to extract shared and private semantic information, respectively, and hierarchical contrastive learning loss function is introduced to enhance features separability: on the one hand, the embedding distance of the two types of features is expanded by cross-view negative sample comparison, and on the other hand, the semantic consistency of similar samples is constrained by using supervised labels. A multi-view shared feature discrimination mechanism is further proposed to strengthen the aggregation of consistent information, and the labels prediction is optimized by a rare-label-oriented decision level fusion strategy. Compared with other state-of-the-art methods, our method shows competitive experimental results on five widely used multi-view multi-label datasets.

## 1 Introduction

In real world, an object can be described as a sample using a single form of data representation or a single set of features Huang et al. (2016). However, a single representation often tends to make it easy to overlook certain hidden features of an object. Many existing studies have begun to focus on analyzing an object or image from different perspectives. Multi-view representations can construct a more complex feature learning space, which facilitates the modeling of complex scenes in the real world Tan et al. (2024). Meanwhile, the category labels of an object requires multiple considerations of different features of the instance for different category classifications, which avoids the phenomenon of category mutual exclusion caused by single-label classification Lin et al. (2020),Tan et al. (2019). Therefore, the Multi-View Multi-Label Classification (MvMLC) task make supervised learning more compatible with real scenarios and are widely studied by scholars nowadays Ou et al. (2024), Boutell et al. (2004). For example, Zhang et al. and Liu et al. utilize matrix decomposition to build the LSA-MML Zhang et al. (2018) and M2LD Liu et al. (2023e) models, which align different attempts of data in the kernel space or build a subspace of views highly correlated with labels. Another example is the CDMM model proposed by Zhao et al. which is based on a nonlinear kernel mapping function approach and introduces the HSIC quasi-metric to obtain consistent labels in all views Zhao et al. (2021).

Currently, the MvMLC task learning focuses on how to effectively fuse these heterogeneous features while connecting the features with all the relevant labels. The current MvMLC tasks are divided into two categories based on different fusion strategies Wei et al. (2025): feature fusion strategies and decision fusion strategies. Feature fusion based approaches classify multi-view features by fusing them into a unified feature representation Zhang et al. (2020). For example, Liu et al. proposed a model of ELSMML, which use dimensionality reduction techniques and stream regularization terms

approximate the low-dimensional structure space to the original data structure and construct unified classifiers Liu et al. (2023a). However, the method tends to focus too much on the consistency of views and lose the specific labels of certain views. In contrast, decision fusion-based methods train separate multi-label classifiers for each view, and averaging or weighted combination of the prediction results from all classifiers . For example, Liu et al. utilizes the confidence generated by the joint attention mechanism to dynamically weight the classification results of each view for fusion Liu et al. (2024).

However, in non-ideal situations, it is often the case that both the view and the label of a sample are incomplete, which poses a significant challenge to the multi-view multi-label task. The main existing approaches to DiMvMLC task are masking the missing views or recovering them in the embedded feature space Li et al. (2024). For the case where both views and labels are missing, Li et al. proposed the NAIM3L model, which relies on the low-rank assumption of the sub-label matrix and uses sub-class correlation to model the global structure of multi-label as a high-rank representation, thereby solving the double incompleteness problem Li & Chen (2021). The DIMC Wen et al. (2023), LMVCAT Liu et al. (2023d) and DICNet Liu et al. (2023c) of deep learning models for DiMVMLC have advantages over traditional learning methods. DIMC and LMVCAT models extract high-level semantic features for multi-view using self-encoder and transformer, respectively. However, the models do not deeply consider the consistency between multi-view. Therefore, DICNet model uses instance-level contrastive learning to guide the encoder to extract high-level semantic features of multi-view to obtain consistent discriminative representations. In addition, Wang et al. proposed using constructing instance similarity matrices using incomplete positive label consistency to compensate for missing views using similar instances Wang et al. (2025). However, most of these methods construct uniform classifiers for all labels, treating all labels equally, but ignore the crucial label imbalance problem in multi-label learning Tan et al. (2019).

To address common problems in DiMvMLC, we propose a classification model based on Rare Label Oriented Discriminative Driven Feature construction methods(RLOD-net). The model consists of four main modules: dual feature extraction framework, hierarchical contrastive loss, discriminative driven unified feature construction, and rare-label-oriented decision level fusion. We map multi-view raw data to shared and private information embedding spaces to simultaneously satisfy the requirements of multi-view consistency and view uniqueness. A hierarchical contrastive learning strategy is utilized to facilitate the distinction and deep extraction of shared and private information. To facilitate the aggregation of multi-view shared information, we employ a shared feature discrimination mechanism to improve the high-level semantic characterization of shared information. Although the shared and private information is available, whether it can be sufficiently fused for proper classification is a great challenge.Therefore, we incorporate the shared information into the private information, highlighting the shared information while preserving the private information to fully integrate two types of information for accurate classification. Finally, the prediction results of all views are output and fused at the decision level to improve the prediction of rare labels. Overall, our proposed model has the following contributions:

1. We propose a novel dual decoupled structure of discriminative driven unified feature construction. It uses a shared discriminant module to strengthen shared semantic information and guide it into private information. While retaining the common information of multi-view, it highlights the characteristics of each view and fully integrates the two types of features.

2. We propose a hierarchical contrastive learning strategy method that considers the feature correlation within the sample and introduces the label space structure to guide the learning of common information between samples. This effectively distinguishes the two types of information and promotes information exchange across samples while retaining the common structure of multi-view.

3. Different from the existing multi-view classification strategy, we combine the private view feature extraction and dynamic decision level fusion strategy, comprehensively consider the private information of each view, and can better capture the personalized label knowledge of the view and achieve accurate classification of rare labels.

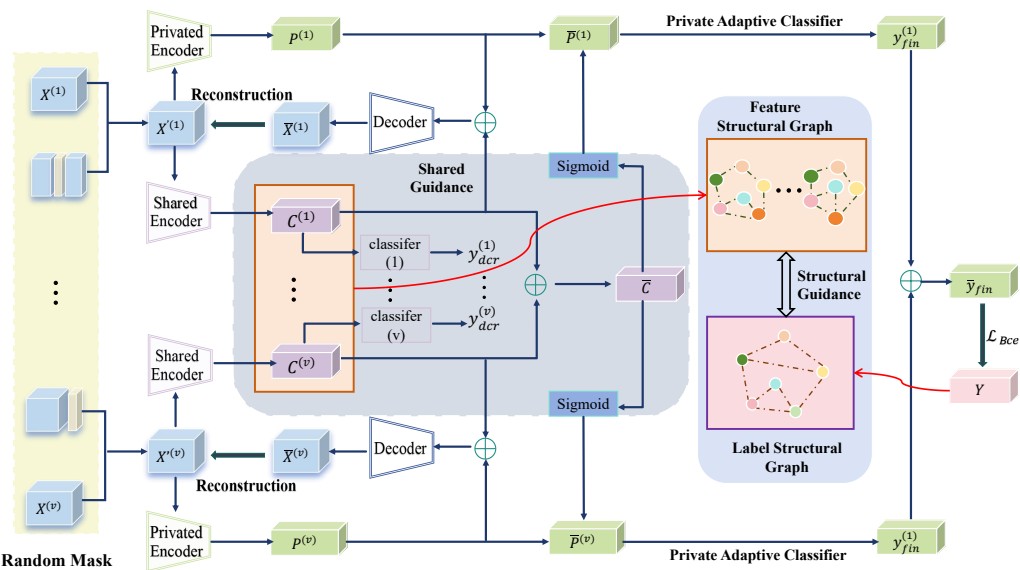

Figure 1: The main procedural framework of RLOD-net includes four parts: dual feature extraction framework, hierarchical contrastive loss, discriminative driven unified feature construction and rare-label-oriented decision level fusion.

## 2 METHODOLOGY

### 2.1 PROBLEM DEFINITION

In this section, we define the DiMvMLC task as follows: Given multi-view multi-label data $\{\mathbf{X}^{(v)} \in \mathbb{R}^{n \times d_v}\}_{v=1}^m$, where $\mathbf{X}^{(v)}$ is the feature matrix of the $v$-th view, $n$ denotes the total number of samples, $d_v$ denotes the $v$-th view feature dimension, and $m$ denotes the total number of views. Each sample contains $v$ views and the dimensions of the $v$-th view are all $d_v$. We define $\mathrm{X}_i^{(v)}$ represented as the $v$-th view of the $i$-th instance sample. Next, given $\mathbf{Y} \in \{0,1\}^{n \times c}$ as the label index matrix, where $n$ denotes the number of samples and $c$ denotes the total number of labels, each sample is multi-label data. $\mathbf{Y}_{i,j} = 1$ indicates that the $i$-th sample belongs to the $j$-th class, and otherwise $\mathbf{Y}_{i,j} = 0$. Since we are targeting incomplete views and labels, for missing views, $\mathbf{W} \in \{0,1\}^{n \times v}$ is introduced as the missing view index matrix, and $\mathbf{W}_{i,j} = 1$ is defined to indicate that the $j$-th view of the $i$-th instance sample is available, and otherwise $\mathbf{W}_{i,j} = 0$. For missing labels, define $\mathbf{G} \in \{0,1\}^{n \times c}$ as the missing label index matrix, where $\mathbf{W}_{i,j} = 1$ denotes the presence of the $j$-th label for the $i$-th sample, and otherwise $\mathbf{W}_{i,j} = 0$.

### 2.2 DUAL FEATURE EXTRACTION FRAMEWORK

In the multi-view domain, multi-view data usually contains two kinds of key information: public semantic information across all views, and view-private information unique to each view. Existing studies have shown that if the presence of these two types of information is fully considered in the modeling process and the complementarity between each view is preserved while obtaining a consistent representation of the multi-view information, the classification ability of the model will be significantly improved Liu et al. (2023b).For this purpose, we use two different stacked encoders $\{\mathrm{E}_v^c : \mathbf{X}'^{(v)} \to \mathbf{C}^{(v)}\}_{v=1}^m$ and $\{\mathrm{E}_v^p : \mathbf{X}'^{(v)} \to \mathbf{P}^{(v)}\}_{v=1}^m$ as shared and private view feature extraction encoders, respectively. The shared feature matrix $\mathbf{C}^{(v)}$ and private feature matrix $\mathbf{P}^{(v)}$ of the $v$-th view are finally obtained.

Subsequently, we assign each view a feature reconstruction decoder $\{\mathrm{D}_v : (\mathbf{P}^{(v)} + \mathbf{C}^{(v)}) \to \bar{\mathbf{X}}^{(v)}\}_{v=1}^m$ to enhance the feature extraction capability of the encoder. The reconstructed data $\bar{\mathbf{X}}^{(v)}$

for the $v$-th view is obtained through the decoder network $D_v$. The missing view index matrix $\mathbf{W}_{i,v}$ is also introduced to eliminate the negative impact of missing views on model training. Finally, the following multi-view raw data reconstruction loss is obtained:

$$\mathcal{L}_{re} = \frac{1}{n} \sum_{v=1}^{m} \sum_{i=1}^{n} \frac{1}{d_v} \|\mathbf{X}_{i:}^v - \bar{\mathbf{X}}_{i:}^v\|_2^2 \mathbf{W}_{i,v} \tag{1}$$

In order to enable the encoder to learn more global and task-relevant feature information from incomplete inputs. Inspired by the masked auto-encoder (MAE) of He et al. (2022), we employ a masking operation to initially mask the initial multi-view vector data, destroying parts of the input data, forcing the encoder to learn a more generalized representation of the features. Specifically, we randomly generate a mask matrix $\mathbf{M}^{(v)} \in \mathbb{R}^{n \times d_v}$ with the same shape and size as the original vector data $\mathbf{X}^{(v)}$ during the training process, where the element values are all 1. Assign a value of 0 to each row of $\mathbf{M}^{(v)}$ at position $[st, st + l]$, where the parameters for each line $st$ is randomly generated by the model. At the same time satisfies that $st$ is smaller than $d_v - \alpha \times d_v$ , the length $l = \alpha \times d_v$, and $\alpha$ are hyperparameters. The original mask data $\mathrm{X}'^{(\mathrm{v})}$ is finally obtained:

$$\{\mathbf{X}'^{(v)}\}_{v=1}^m = \{\mathbf{X}^{(v)} \odot \mathbf{M}\}_{v=1}^m \tag{2}$$

where $\odot$ denotes element-wise multiplication.

## 2.3 HIERARCHICAL CONTRASTIVE LOSS

In order to ensure that the shared embedding features of the same sample on different views remain consistent in the latent space and to effectively distinguish shared view features from private view features, we design a hierarchical contrastive loss function. This loss function integrates instance-level contrastive learning loss and label-level contrastive learning loss.

### 2.3.1 INSTANCE-LEVEL CONTRASTIVE LOSS

In the potential feature space learned by the autocoder, we learn by contrastive learning to maximize the consistency between different views. It assumes that each sample is an independent instance and pushes negative samples farther away by bringing positive sample pairs closer together Logeswaran & Lee (2018),Oord et al. (2018). The loss function for general contrast learning can be expressed as:

$$- \log \frac{e^{\mathcal{S}(x,x^+)}}{e^{\mathcal{S}(x,x^+)} + \sum_{x^- \in \mathcal{F}(x)} e^{\mathcal{S}(x,x^-)}}$$

$$\approx -\mathcal{S}(x, x^+) + \log \sum_{x^- \in \mathcal{F}(x)} e^{\mathcal{S}(x,x^-)} \tag{3}$$

Specifically, in 2.2 it is required that the shared feature encoder tries to mine the common features between views, preserving the basic attributes of the samples held by all views. So we make $c_i^{(v)}$ and $c_i^{(u)}$ as positive sample pairs and finally get the positive sample similarity $\mathcal{S}(c_i^{(v)}, c_i^{(u)})$ , where $u \neq v$ and $u, v \in m$. Get the positive sample pair loss for the $i$-th instance:

$$\mathcal{L}_{pos} = \left[ \left( \sum_{v \in f}^{m} \sum_{u \neq v, u \in f}^{m} \left( \mathcal{S}(c_i^{(v)}, c_i^{(u)}) + 1 \right) / 2 \right) \right]_{\text{mean}} \tag{4}$$

where $f = \{(i,v) | \mathbf{W}_{i,v} = 1\}$ indicates the set of indexes at which the sample view exists, and $\mathbf{W}_{i,v} = 1$ indicates the $v$-th view of the $i$-th sample exists.

To disentangle shared and private features, we designate negative samples as $\mathcal{S}(c_i^{(v)} , p_i^{(u)})$. We treat the private information from different views as negative sample pairs, which can be formulated as $\mathcal{S}(p_i^{(v)} , p_i^{(u)})$, to ensure that the private feature from multi-view of the same sample exhibits uniqueness and discriminative power. Get the negative sample pair loss function for the $i$-th instance:

$$\mathcal{L}_{neg} = [2 \sum_{v=1}^{m} \sum_{u=1}^{m} \mathcal{S}(c_i^{(v)}, p_i^{(u)})^2 + \sum_{v=1}^{m} \sum_{u \neq v}^{m} \mathcal{S}(p_i^{(v)}, p_i^{(u)})^2]_{mean} \tag{5}$$

We use cosine similarity to calculate the similarity between two samples, denoted as $\mathcal{S}(x, x^{+/-}) = \frac{x^T x^{+/-}}{\|x\|_2 \|x^{+/-}\|_2}$. Therefore, according to Eq.(3), the final instance-level contrastive loss function can be formulated as:

$$\mathcal{L}_{ic} = -\mathcal{L}_{pos} + \mathcal{L}_{neg} \tag{6}$$

### 2.3.2 LABEL-LEVEL CONTRASTIVE LOSS

To taking full account of the correlation between samples of different labels, we introduce a priori label correlation to guide the shared view features of similar instances to approach each other. Therefore, we define the similarity matrix between samples in the incomplete multi-view space as $\mathbf{L} = \big((\mathbf{Y} \odot \mathbf{G})(\mathbf{Y} \odot \mathbf{G})^T\big) \cdot / \big(\mathbf{G}\mathbf{G}^T\big)$ and $\mathbf{L} \in [0,1]^{n \times n}$. Furthermore, we define the correlation matrix of multi-view as:

$$\mathbf{F}_{i,j}^{(v)} = \frac{c_i^{(v)^T} c_j^{(v)}}{\left\|c_i^{(v)}\right\|_2 \left\|c_j^{(v)}\right\|_2} \tag{7}$$

Further, we facilitate the learning of shared information about similar labels by aligning the spatial structure between sample features with the spatial structure of labeled embeddings through similarity metric learning. A simple and effective label comparison loss can be expressed as:

$$\mathcal{L}_{lc} = -\sum_{v=1}^{m} \frac{1}{d_v N} \sum_{i \in f}^{n} \sum_{j \in f, i \neq j}^{n} \left\| \mathbf{F}_{i,j}^{(v)} - \mathbf{L}_{i,j} \right\|_2^2 \tag{8}$$

where $N = \sum_{i,j} \mathbf{W}_{i,v} \mathbf{W}_{j,v}$ denote the number of valid instance pairs. Through the $l_2$-norm, the labels embedding information $\mathbf{L}_{i,j}$ guides the learning of $\mathbf{F}_{i,j}^{(v)}$, such that the label relevance within the shared information space progressively approximates the similarity of the true labels.

In contrast to traditional contrastive learning methods, we instead construct a similarity matrix for the shared information across multiple views and design a loss function for feature correlation and label correlation based on $l_2$-norm. This ensures that the spatial structure among samples is consistent with the geometric structure among sample labels, which not only preserves the inherent characteristics of multi-view data but also strengthens the structural correlation between features and labels through the constraints of the loss function.

In summary, the final hierarchical contrastive loss is formulated as:

$$\mathcal{L}_{hc} = \alpha \mathcal{L}_{ic} + \beta \mathcal{L}_{lc} \tag{9}$$

where $\alpha, \beta$ represents the corresponding non-negative penalty parameter.

## 2.4 DISCRIMINATIVE DRIVEN UNIFIED FEATURE CONSTRUCTION

After capturing the shared information across multi-view, we aim to generate a unified representation with the commonalities and complementarities of multi-view. To address this,we introduce a discriminative driven unified feature construction approach to capture a consistent representation of shared and private information across different views.

Firstly, we perform a shared feature discrimination operation to fully aggregate multi-view consensus information. Specifically, we assign a classifier to each view, obtaining preliminary classification results $\mathbf{y}_{dcr}^{(v)}$ for each view. The following multi-view discriminative loss can be obtained:

$$\mathcal{L}_{dcr} = \frac{1}{n} \sum_{i=1}^{n} \sum_{v=1}^{m} \left[ \sum_{j=1}^{c} \left[ \mathbf{Y}_{i,j} \log \mathbf{y}_{i,j}^{(v)} + (1 - \mathbf{Y}_{i,j}) \log(1 - \mathbf{y}_{i,j}^{(v)}) \right] G_{i,j} \right]_{\text{mean}} \tag{10}$$

where $\mathbf{Y}_{i,j}$ denotes the $j$-th ground truth label for the $i$-th sample, while $\mathbf{y}_{i,j}^{(v)}$ represents the predicted label for the $v$-th view of the same sample. We introduce the label missing index matrix $\mathbf{G}$ to mitigate the impact of missing labels on the pre-classification process. By minimizing this loss function, we aim to align the classification outcomes of each view more closely with the ground truth labels.

This approach guides the encoder in acquiring more effective shared features.Ultimately, multi-view fused shared features are obtained:

$$\bar{\mathbf{C}}_i = \frac{\sum_{v=1}^m \mathbf{C}_i^v \mathbf{W}_{i,v}}{\Sigma_{v=1}^m \mathbf{W}_{i,v}} \tag{11}$$

Next, we merge the shared features and private features in the following way:

$$\bar{\mathbf{P}}_i^{(v)} = sig(\bar{\mathbf{C}}_i)\mathbf{P}_i^{(v)} \tag{12}$$

where $sig(\cdot)$ denotes the sigmoid activation function. The shared information embedding features $\overline{\mathbf{C}}$, processed through a sigmoid activation function, interact with the private information embedding features $\mathbf{P}^{(v)}$ of each view. Sigmoid nonlinear features enables the nonlinear relationship between shared and proprietary features to be captured, allowing the respective characteristics of information to be retained more effectively during fusion. Compared with the previous approach of integrating private information into shared information, we approach facilitates the highlighting of multi-view shared labeling information within private information while preserving the distinctiveness of the original view.

## 2.5 Rare-Label-Oriented Decision Level Fusion

In most cases, many models use feature level fusion to learn a unified classifier. However the approach tends to lose some view private or rare label information. Therefore, we perform feature fusion at the decision level to focus on rare but equally important labels that are specific to certain views. A fully connected layer is assigned to each view and the view fusion private features are classified:

$$\mathbf{y}_{fin}^{(v)} = \bar{\mathbf{P}}_i^{(v)}\mathcal{W}^{(v)} + \mathcal{B}^{(v)} \tag{13}$$

where $\mathcal{W}^{(v)} \in \mathbb{R}^{n \times d_v}$ and $\mathcal{B}^{(v)} \in \mathbb{R}^{n \times c}$ are learnable parameters of classifiers. We obtain the final predicted labels according to the following decision level feature fusion strategy:

$$\bar{\mathbf{y}}_{fin} = \frac{\sum_{v=1}^m \mathbf{y}_{fin}^{(v)}}{\sum_{v=1}^m \mathbf{W}_{i,v}} \tag{14}$$

Finally, the binary cross entropy loss is used to calculate the classification loss for decision fusion as follows:

$$\mathcal{L}_{df} = \frac{1}{n}\sum_{i=1}^n \left[ \sum_{j=1}^c \left[ \mathbf{Y}_{i,j}\log\bar{\mathbf{y}}_{i,j} \quad + (1-\mathbf{Y}_{i,j})\log(1-\bar{\mathbf{y}}_{i,j}) \right]\mathbf{G}_{i,j} \right] \tag{15}$$

Combining the above reconstruction loss $\mathcal{L}_{re}$, hierarchical contrastive loss $\mathcal{L}_{hc}$, multi-view discrimination loss $\mathcal{L}_{dcr}$, and final decision fusion $\mathcal{L}_{df}$, our total loss equation is:

$$\mathcal{L} = \gamma\mathcal{L}_{re} + \mathcal{L}_{hc} + \mathcal{L}_{dcr} + \mathcal{L}_{df} \tag{16}$$

where $\gamma$ is the corresponding non-negative penalty parameter. We show the detailed training process in Algorithm 1.

## 3 Experiments

### 3.1 Experiment initial setup

#### 3.1.1 Datasets

In this paper, we refer to Tan et al. (2018),Duan et al. (2025) and select five common multi-view multi-label datasets to test the training capability of the model. They are Corel5k, Pascal07, ESPGame, Iaprtc12, and Mirflickr. All of the above datasets consist of 6 views and each sample contains multi-label, which meets the dataset requirements needed for multi-view multi-label task Li & Chen (2021).This experiment is for double incomplete multi-view multi-label data, and the dataset is treated as follows: (1) 70% of the samples are randomly selected as the training set, 15% as the validation set, and 15% as the test set. (2) Perform 50% random masking of each view and label for all samples to ensure that at least one view and label is retained for training for each sample, and define the missing view and label index matrix. Note that the labels of the validation and test sets are intact and are not processed in any way.

---

**Algorithm 1** Training process of SGPA-net

---

**Input**: Incomplete multi-view data $\{\mathbf{X}^{(v)}\}_{v=1}^m$ with missing-view indicator matrix $\mathbf{W} \in \{0,1\}^{n \times v}$, and corresponding multi-label matrix $\mathbf{Y} \in \{0,1\}^{n \times c}$ with missing-label indicator matrix $\mathbf{G} \in \{0,1\}^{n \times c}$.
**Parameter**: $\alpha, \beta, \gamma$, learning rate, and training epochs.
**Output**: Prediction results $\bar{\mathbf{y}}_{fin}$.

1: Initialize model parameters and set hyperparameters.
2: $t = 0$.
3: **while** $t < e$ **do**
4:    Construct random feature mask matrices $\{\mathbf{M}^{(v)}\}_{v=1}^m$.
5:    Compute masked input data $\{\mathbf{X}^{'(v)}\}_{v=1}^m$ by Eq. (2).
6:    Obtain common information $\{\mathbf{C}^{(v)}\}_{v=1}^m$ and proprietary information $\{\mathbf{P}^{(v)}\}_{v=1}^m$ .
7:    Obtain $\mathcal{L}_{ic}$ and $\mathcal{L}_{lc}$ by Eq. (6)and Eq. (8), respectively. And obtain $\mathcal{L}_{re}$ by Eq. (1).
8:    Obtain the result $\mathbf{y}_{dcr}^{(v)}$ and $\mathcal{L}_{dcr}$ by Eq. (10).
9:    Obtain fused common information $\bar{\mathbf{C}}$ and fused representation $\bar{\mathbf{P}}$ by Eq. (11) and by Eq. (12).
10:   Get the categorization results for each view $\mathbf{y}_{fin}^{(v)}$.
11:   Get the multi-view fusion classification result $\bar{\mathbf{y}}_{fin}$ by Eq. (13) and obtain $\mathcal{L}_{df}$ by Eq. (14).
12:   Compute total loss $\mathcal{L}$ by Eq. (16).
13:   Update network parameters.
14:   $t = t + 1$.
15: **end while**

---

### 3.1.2 COMPARISON METHODS

We selected eight of the latest and most representative methods for comparison. These include NAIM3L Li & Chen (2021), CDMM Zhao et al. (2021), DIMC Wen et al. (2023), LMVCAT Liu et al. (2023d), DICNet Liu et al. (2023c), MTD Liu et al. (2023b), UPDGD-net Xie et al. (2024) and CSCR Wang et al. (2025). In addition to the MTD and UPDGD-net model, other models have been described in previous relevant sections. MTD model is a two-channel decoupled framework based classification model that uses global label correlation to guide graph regularization. UPDGD-net imposes dyadic graph constraints on multi-view embedded features and utilizes uncertainty-aware pseudo-tagging strategies to fill in missing labels. It is worth noting that the CDMM model is a training model for handling the complete multi-view multi-label task, so the average of the view features is used to fill in the missing views and the missing labels are set to '0'. In this experiment, all the models select the best experimental parameters labeled in the paper to obtain the results after several experiments. Our model has three hyperparameters $\alpha$, $\beta$ and $\gamma$, which are set to 0.001, 0.1 and 0.1, respectively.

### 3.1.3 EVALUATION METRICS

Referring to Liu et al. (2023b),Zhao et al. (2022), we select common DiMvMLC evaluation metrics, including: Average Precision (AP), Ranking Loss (RL), Adapted Area Under Curve (AUC), OneError (OE) and coverage (Cov). For AP and AUC the higher the value, the better the performance. For RL, OE and Cov, the lower the value, the better the performance. Therefore, we set "1-RL", "1-OE" and "1-Cov", expecting that all evaluation metrics are as large as possible.

## 3.2 ANALYSIS OF EXPERIMENTAL RESULTS

As shown in Table 1, we give the performance of the model on five datasets, with 50% missing view rate, and 50% missing label rate, and show the comparison data with eight other models. All experiments were repeated several times to ensure accuracy and statistical significance and to avoid chance. According to the results in the table, we make the following observations:

1. Compared to the other eight methods, our proposed method shows superior classification performance on all datasets, ranking first in most of the metrics. Among them, in terms

Table 1: Experimental results of nine methods on the five databases with 50% missing-view rate, 50% missing-label rate, and 70% training samples.

| Dataset | Metric | NAIM3L | CDMM | DIMC | LMVCAT | DICNet | MTD | UPDGD-net | CSCR | OURS |
|---|---|---|---|---|---|---|---|---|---|---|
| Corel5k | AP | $0.309_{0.004}$ | $0.289_{0.004}$ | $0.351_{0.007}$ | $0.377_{0.008}$ | $0.381_{0.017}$ | $0.414_{0.006}$ | $0.411_{0.013}$ | $0.386_{0.014}$ | $\mathbf{0.432}_{0.007}$ |
| | 1-RL | $0.878_{0.002}$ | $0.768_{0.0048}$ | $0.863_{0.004}$ | $0.877_{0.005}$ | $0.879_{0.005}$ | $0.892_{0.002}$ | $0.902_{0.003}$ | $0.888_{0.007}$ | $\mathbf{0.909}_{0.002}$ |
| | AUC | $0.881_{0.002}$ | $0.768_{0.0041}$ | $0.866_{0.004}$ | $0.880_{0.006}$ | $0.882_{0.004}$ | $0.895_{0.002}$ | $0.905_{0.003}$ | $0.891_{0.008}$ | $\mathbf{0.911}_{0.002}$ |
| | 1-OE | $0.350_{0.009}$ | $0.337_{0.006}$ | $0.426_{0.014}$ | $0.444_{0.014}$ | $0.472_{0.027}$ | $0.491_{0.008}$ | $0.479_{0.025}$ | $0.457_{0.020}$ | $\mathbf{0.507}_{0.001}$ |
| | 1-Cov | $0.725_{0.005}$ | $0.493_{0.010}$ | $0.685_{0.008}$ | $0.722_{0.012}$ | $0.721_{0.011}$ | $0.750_{0.006}$ | $0.775_{0.004}$ | $0.735_{0.017}$ | $\mathbf{0.784}_{0.004}$ |
| Iaprtc12 | AP | $0.261_{0.001}$ | $0.240_{0.003}$ | $0.299_{0.004}$ | $0.316_{0.004}$ | $0.325_{0.005}$ | $0.333_{0.004}$ | $0.331_{0.005}$ | $0.308_{0.005}$ | $\mathbf{0.349}_{0.004}$ |
| | 1-RL | $0.848_{0.001}$ | $0.752_{0.004}$ | $0.852_{0.003}$ | $0.869_{0.003}$ | $0.873_{0.001}$ | $0.876_{0.001}$ | $0.882_{0.002}$ | $0.863_{0.004}$ | $\mathbf{0.887}_{0.001}$ |
| | AUC | $0.850_{0.001}$ | $0.751_{0.003}$ | $0.855_{0.003}$ | $0.871_{0.002}$ | $0.874_{0.002}$ | $0.877_{0.001}$ | $0.883_{0.002}$ | $0.845_{0.003}$ | $\mathbf{0.888}_{0.001}$ |
| | 1-OE | $0.390_{0.005}$ | $0.359_{0.003}$ | $0.432_{0.007}$ | $0.437_{0.005}$ | $0.465_{0.007}$ | $0.475_{0.008}$ | $0.457_{0.008}$ | $0.447_{0.009}$ | $\mathbf{0.484}_{0.007}$ |
| | 1-Cov | $0.592_{0.004}$ | $0.357_{0.004}$ | $0.589_{0.006}$ | $0.645_{0.005}$ | $0.649_{0.004}$ | $0.650_{0.003}$ | $0.679_{0.003}$ | $0.602_{0.013}$ | $\mathbf{0.680}_{0.004}$ |
| Espgame | AP | $0.246_{0.002}$ | $0.235_{0.003}$ | $0.281_{0.005}$ | $0.293_{0.004}$ | $0.299_{0.004}$ | $0.306_{0.003}$ | $0.308_{0.004}$ | $0.304_{0.004}$ | $\mathbf{0.318}_{0.004}$ |
| | 1-RL | $0.818_{0.002}$ | $0.721_{0.0032}$ | $0.814_{0.0026}$ | $0.828_{0.002}$ | $0.833_{0.002}$ | $0.837_{0.001}$ | $0.843_{0.003}$ | $0.840_{0.003}$ | $\mathbf{0.848}_{0.001}$ |
| | AUC | $0.824_{0.002}$ | $0.718_{0.003}$ | $0.818_{0.002}$ | $0.833_{0.002}$ | $0.837_{0.002}$ | $0.842_{0.001}$ | $0.848_{0.003}$ | $\mathbf{0.865}_{0.004}$ | $0.853_{0.001}$ |
| | 1-OE | $0.339_{0.003}$ | $0.367_{0.007}$ | $0.416_{0.011}$ | $0.433_{0.010}$ | $0.440_{0.010}$ | $0.450_{0.010}$ | $0.447_{0.007}$ | $0.454_{0.008}$ | $\mathbf{0.459}_{0.010}$ |
| | 1-Cov | $0.571_{0.003}$ | $0.363_{0.003}$ | $0.547_{0.006}$ | $0.590_{0.006}$ | $0.598_{0.006}$ | $0.601_{0.002}$ | $0.622_{0.009}$ | $0.605_{0.005}$ | $\mathbf{0.627}_{0.002}$ |
| Mirflicker | AP | $0.551_{0.002}$ | $0.483_{0.003}$ | $0.587_{0.003}$ | $0.595_{0.005}$ | $0.586_{0.004}$ | $0.608_{0.004}$ | $0.606_{0.005}$ | $0.589_{0.005}$ | $\mathbf{0.615}_{0.004}$ |
| | 1-RL | $0.844_{0.001}$ | $0.794_{0.001}$ | $0.865_{0.002}$ | $0.864_{0.003}$ | $0.861_{0.002}$ | $0.876_{0.002}$ | $0.870_{0.003}$ | $0.866_{0.002}$ | $\mathbf{0.878}_{0.001}$ |
| | AUC | $0.837_{0.001}$ | $0.759_{0.002}$ | $0.852_{0.000}$ | $0.851_{0.002}$ | $0.849_{0.002}$ | $0.862_{0.002}$ | $0.857_{0.002}$ | $0.854_{0.002}$ | $\mathbf{0.864}_{0.002}$ |
| | 1-OE | $0.585_{0.003}$ | $0.507_{0.004}$ | $0.633_{0.006}$ | $0.641_{0.009}$ | $0.638_{0.008}$ | $0.655_{0.002}$ | $0.659_{0.007}$ | $0.643_{0.007}$ | $\mathbf{0.665}_{0.002}$ |
| | 1-Cov | $0.631_{0.002}$ | $0.528_{0.002}$ | $0.654_{0.003}$ | $0.667_{0.003}$ | $0.647_{0.003}$ | $0.678_{0.002}$ | $0.678_{0.002}$ | $0.653_{0.004}$ | $\mathbf{0.682}_{0.004}$ |
| Pascal07 | AP | $0.488_{0.003}$ | $0.450_{0.006}$ | $0.530_{0.007}$ | $0.525_{0.009}$ | $0.505_{0.008}$ | $0.550_{0.005}$ | $0.537_{0.014}$ | $0.521_{0.009}$ | $\mathbf{0.561}_{0.004}$ |
| | 1-RL | $0.783_{0.001}$ | $0.726_{0.003}$ | $0.811_{0.005}$ | $0.812_{0.006}$ | $0.786_{0.004}$ | $0.830_{0.005}$ | $0.825_{0.007}$ | $0.800_{0.006}$ | $\mathbf{0.837}_{0.003}$ |
| | AUC | $0.811_{0.001}$ | $0.729_{0.002}$ | $0.833_{0.004}$ | $0.833_{0.007}$ | $0.809_{0.004}$ | $0.849_{0.003}$ | $0.847_{0.006}$ | $0.822_{0.006}$ | $\mathbf{0.857}_{0.003}$ |
| | 1-OE | $0.421_{0.006}$ | $0.377_{0.007}$ | $0.448_{0.009}$ | $0.430_{0.015}$ | $0.428_{0.015}$ | $0.456_{0.008}$ | $0.432_{0.020}$ | $0.445_{0.016}$ | $\mathbf{0.467}_{0.007}$ |
| | 1-Cov | $0.727_{0.002}$ | $0.664_{0.003}$ | $0.760_{0.006}$ | $0.764_{0.007}$ | $0.732_{0.0034}$ | $0.783_{0.002}$ | $0.781_{0.006}$ | $0.745_{0.007}$ | $\mathbf{0.790}_{0.002}$ |

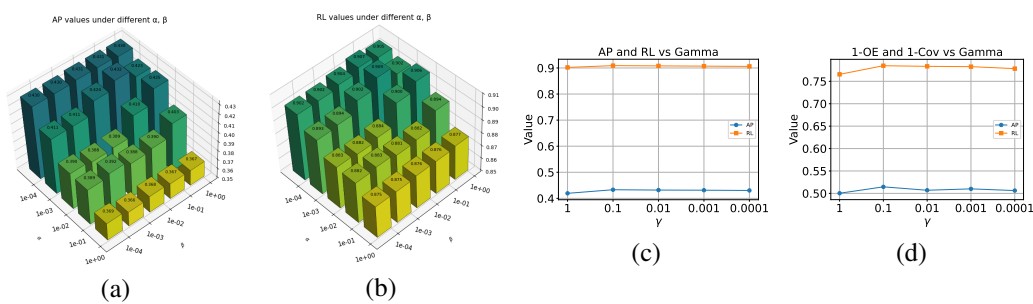

(a)          (b)          (c)          (d)

Figure 2: $\alpha, \beta, \gamma$ Sensitivity testing of Corel5k evaluation indicators:(a) AP values under different $\alpha, \beta$ (b) RL values under different $\alpha, \beta$ (c) AP values under different $\gamma$ (d) RL values under different $\gamma$

of average precision AP, it shows 1% to 2% improvement on all datasets. Meanwhile, our model rankings are ranked first on all datasets.

2. In addition, the top-ranked models, including DICNet, MTD, VCMN, and CSCR, employ contrastive learning to improve the deep representation of multi-view embedded features. This suggests that contrastive learning can help improve the model classification ability to some extent. It introduces cross-view correlation while maintaining consistency among multi-view to obtain a more discriminative view embedding representation.

With reference to Tan et al. (2019), we conducted experiments to quantify the model's ability to classify rare labels.We experimentally quantified the model's ability to classify rare labels, dividing rare labels into three levels (rare1,rare2,rare3) based on the imbalance ratio ($IR_c$).The experimental results are shown in Table 3. The results show that the model outperforms other models in processing rare labels. Furthermore, AP decreases with increasing $IR_c$, and 1-RL changes more slowly,

reflecting that rarity has a non-linear effect on the model's ranking ability.For detailed experimental design, see Appendix A.

In our model, there are three hyperparameters, $\alpha$, $\beta$ and $\gamma$, which need to be set before training. To test the optimal hyperparameter configuration, we chose to conduct experiments on the corel5k dataset with 50% missing views, 50% missing labels and 70% training samples. We chose $\alpha$ and $\beta$ to plot the results regarding both AP and AUC parameters. As shown in Fig.2 (a),(b), we chose $\alpha$ and $\beta$ as [0.001,0.1] and the model achieved satisfactory performance results. And fixing $\alpha$ and $\beta$ as [0.001,0.1], the optimal parameter of $\gamma$ is tested, as shown in Fig. 2 (c), (d), the training results of AP, RL, OE and Cov in the dataset corel5k with different $\gamma$ are plotted, and it can be obtained that the model is optimal when $\gamma$ is 0.1.

### 3.3 ABLATION STUDY

To demonstrate the effectiveness of each part of the model, we performed ablation experiments on Corel5k datasets containing 50% instances, 50% missing labels and 70% training samples. We sequentially removed $\mathcal{L}_{re}$, $\mathcal{L}_{hc}$ and $\mathcal{L}_{dcr}$ and performed controlled experiments. Meanwhile, we fuse the features of each view and perform classification through a unified classifier, and construct a feature-level fusion strategy model for comparative experiments. As shown in the table 2, we tested the effect of each module on the model's classification results. The results in the table show that each component is crucial to the enhancement of the model classification results. And $\mathcal{L}_{dcr}$ has the most significant effect on model enhancement. It also confirms that decision level fusion is superior to feature level fusion.

Table 2: AP, AUC, 1-RL results of ablation experiments in the Corel5k and Pascal07 dataset. Dec-fusion and Fea-fusion respectively represent decision level fusion strategy and feature level fusion strategy ,which feature level fusion strategy is to fuse all views first and then perform classification. ($\mathcal{L}_{ic} + \mathcal{L}_{lc} = \mathcal{L}_{hc}$ )

| Method | core15k | | | pascal07 | | |
|---|---|---|---|---|---|---|
| | AP | AUC | 1-RL | 1-OE | 1-Cov | 1-RL |
| Dec-fusion | 0.416 | 0.896 | 0.893 | 0.540 | 0.835 | 0.819 |
| Fea-fusion | 0.409 | 0.890 | 0.893 | 0.533 | 0.832 | 0.817 |
| $\mathcal{L}_{ic}$ | 0.418 | 0.896 | 0.897 | 0.546 | 0.840 | 0.816 |
| $\mathcal{L}_{lc}$ | 0.422 | 0.901 | 0.896 | 0.544 | 0.839 | 0.822 |
| $\mathcal{L}_{ic} + \mathcal{L}_{lc}$ | 0.425 | 0.901 | 0.898 | 0.549 | 0.843 | 0.824 |
| $\mathcal{L}_{dcr}$ | 0.427 | 0.904 | 0.901 | 0.551 | 0.849 | 0.827 |
| $\mathcal{L}_{re}$ | 0.418 | 0.899 | 0.896 | 0.547 | 0.842 | 0.823 |
| $\mathcal{L}_{hc} + \mathcal{L}_{dcr}$ | 0.430 | 0.910 | 0.907 | 0.560 | 0.855 | 0.836 |
| $\mathcal{L}_{hc} + \mathcal{L}_{re}$ | 0.425 | 0.902 | 0.900 | 0.555 | 0.852 | 0.832 |
| $\mathcal{L}_{dcr} + \mathcal{L}_{re}$ | 0.428 | 0.905 | 0.905 | 0.559 | 0.853 | 0.834 |
| all | **0.432** | **0.911** | **0.909** | **0.561** | **0.857** | **0.837** |

## 4 CONCLUSION

In this paper, we propose a multi-view multi-label classification model based on rare-label-oriented discriminative driven feature construction method. Existing methods usually rely on multi-view consistency information for feature fusion, but single consensus-oriented modeling weakens the ability of view specificity to characterize rare labels. In this study, we design a dual feature fusion mechanism: firstly, we construct mutli-views discrimination module to enhance the aggregation of shared semantics through cross-view consistency constraints; secondly, we proposed a dynamic fusion strategy at the decision layer that injects shared semantics into private features in the form of attention weights, while highlighting the discriminative nature of each view's private labels, to achieve equal classification for all labels.We compare our model with several frontier methods on common datasets and perform modular ablation experiments, strongly demonstrating the superior performance of our model. The specific code can be found in the supplementary material.

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

## A  EVALUATION OF RARE LABEL CLASSIFICATION PERFORMANCE

With reference to Tan et al. (2019), we conducted experiments to quantify the model's ability to classify rare labels. Let $IR_c$ denote the imbalance ratio for label j, calculated as the ratio of the number of negative samples to the number of positive samples for label j. We used $IR_c$ to classify labels into common and rare labels. To validate the model's ability to classify rare labels, we categorized rare labels into three levels: rare1 ( $50 < IR_c \leq 150$ ), rare2 ( $150 < IR_c \leq 250$ ), and rare3 ($IR_c > 250$ ). Finally, the number of rare labels in the three levels was 61 in rare1, 46 in rare2, and 109 in rare3. The experimental results are shown in Table 3. As can be seen from the table, our model outperforms other models in handling rare labels, validating its robustness in classifying rare labels. Furthermore, the AP metric decreases with increasing IRc, indicating that increasing rarity significantly impacts the model's ability to rank and identify positive samples. The value of 1-RL changes more slowly, indicating that it is less affected by changes in label rarity, reflecting that the impact of rarity on the model's ranking ability is not a simple linear relationship.

Table 3: AP and 1-RL experimental results of different rare label cases on the corel5k dataset with 50% missing-view rate, 50% missing-label rate, and 70% training samples

| $IR_c$ | Metric | NAIM3L | CDMM | DIMC | LMVCAT | DICNet-main | MTD | UPDGD | CSCR | OURS |
|---|---|---|---|---|---|---|---|---|---|---|
| rare1 | AP | $0.201_{0.001}$ | $0.187_{0.002}$ | $0.168_{0.001}$ | $0.192_{0.012}$ | $0.213_{0.011}$ | $0.220_{0.003}$ | $0.207_{0.004}$ | $0.202_{0.001}$ | $\mathbf{0.230}_{0.002}$ |
| | 1-RL | $0.870_{0.001}$ | $0.873_{0.001}$ | $0.865_{0.003}$ | $0.887_{0.006}$ | $0.902_{0.006}$ | $0.915_{0.003}$ | $0.901_{0.005}$ | $0.898_{0.001}$ | $\mathbf{0.920}_{0.004}$ |
| rare2 | AP | $0.091_{0.010}$ | $0.093_{0.001}$ | $0.057_{0.001}$ | $0.071_{0.007}$ | $0.084_{0.010}$ | $0.085_{0.005}$ | $0.087_{0.004}$ | $0.078_{0.005}$ | $\mathbf{0.094}_{0.006}$ |
| | 1-RL | $0.829_{0.004}$ | $0.833_{0.005}$ | $0.939_{0.002}$ | $0.949_{0.003}$ | $0.961_{0.003}$ | $0.961_{0.004}$ | $0.959_{0.005}$ | $0.954_{0.001}$ | $\mathbf{0.967}_{0.003}$ |
| rare3 | AP | $0.040_{0.001}$ | $0.045_{0.007}$ | $0.022_{0.001}$ | $0.037_{0.005}$ | $0.036_{0.005}$ | $0.043_{0.003}$ | $0.035_{0.003}$ | $0.033_{0.001}$ | $\mathbf{0.048}_{0.002}$ |
| | 1-RL | $0.819_{0.001}$ | $0.729_{0.002}$ | $0.924_{0.001}$ | $0.937_{0.005}$ | $0.943_{0.005}$ | $0.940_{0.002}$ | $0.949_{0.004}$ | $0.940_{0.001}$ | $\mathbf{0.946}_{0.003}$ |

## B  LEARNING CURVE

As in Figure 3, we plot the curves of average accuracy(AP) and model loss with the number of training rounds.The AP shows an upward trend with the number of training rounds. Except for the dataset other than corel5k, the other four datasets all stabilize after 75 training rounds in terms of the AP parameter, while corel5k stabilizes only after 150 training rounds. And the training loss of the model shows a decreasing trend with the number of training rounds, but compared with the first 150 rounds drop, the model loss in the last 50 rounds has a weaker impact on the performance evaluation index, so the optimal number of training rounds for our model should be set between 150 and 200 rounds.

## C  DIFFERENT MISSING PERCENTAGES FOR VIEWS OR LABELS

As shown in Figure 4, we compare the training results with different label loss rates at 50% view loss rate and the learning results with different view loss rates at 50% label loss rate, respectively. From the figure, it can be seen that all performance metrics show a decreasing trend as the label loss rate and view loss rate increase. It is obvious that missing views have a greater impact on the classification task and the performance metrics decrease more significantly. One possible reason is that the absence of views results in fewer effective features, which reduces the model's classification performance. However, the absence of labels can compensate for the effect of insufficient supervision due to the absence of labels by enabling cross-sample information exchange through the category relevance of the two-layer comparison learning module.

## D  DATASET

## E  EVALUATION METRICS

We select common DiMvMLC evaluation metrics, including: Average Precision (AP), Ranking Loss (RL), Adapted Area Under Curve (AUC), OneError (OE) and Coverage (Cov).

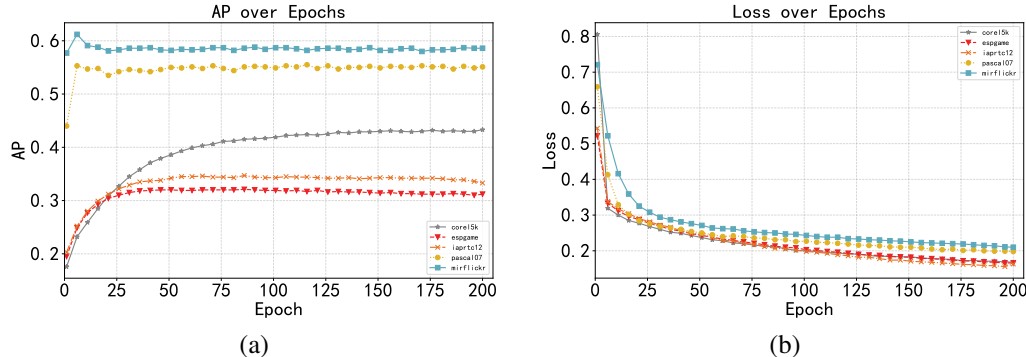

Figure 3: Variation of experimental parameters with the number of training rounds:(a) AP variation with epochs (b)Loss variation with epochs.

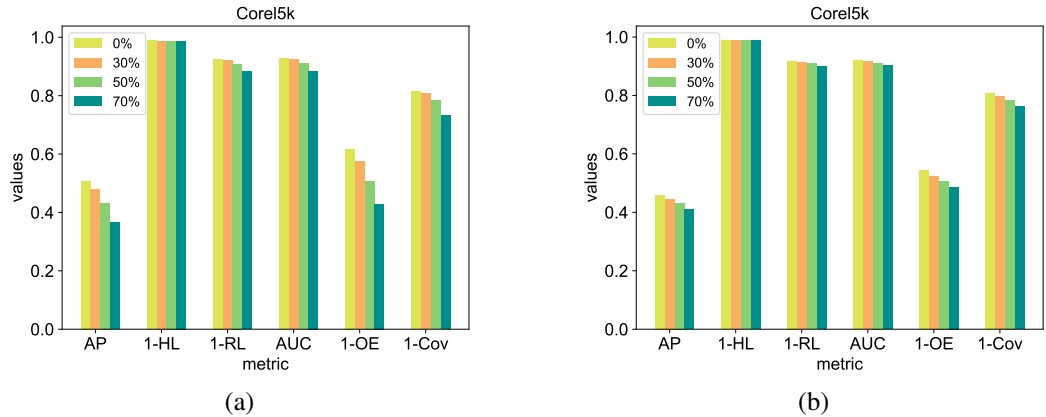

Figure 4: Results on the Corel5k dataset with (a) different missing-view rates, (b) different missing-label rates.

- Average Precision(AP): is used to evaluate the model's ability to rank each label and prediction accuracy, i.e., the model's ability to rank positive labels first and its accuracy in recognizing positive labels.

- Ranking Loss (RL): calculates the model's ability to rank positive and negative labels, which calculates the probability that a positive label is ranked behind a negative label. The smaller the sorting loss, the stronger the model's ability to sort positive labels, i.e., positive labels are more likely to be ranked in front of negative labels.

- Adapted Area Under Curve(AUC): indicates the ability of the model to randomly select positive and negative samples for differentiation, the closer the AUC value is to 1, the better the model performs. the closer the AUC is to 1, the better the model's classification performance is.

- OneError (OE): evaluates whether the model can correctly predict the label with the highest confidence level on each sample. It calculates the proportion of the highest confidence label

Table 4: Attributes of five common datasets

| dataset | view | sample | category | train sample |
|---|---|---|---|---|
| corel5k | 6 | 4999 | 260 | 3500 |
| Pascal07 | 6 | 9963 | 20 | 6975 |
| ESPGame | 6 | 20770 | 268 | 14539 |
| Iaprtc12 | 6 | 19627 | 291 | 13739 |
| Mirflickr | 6 | 25000 | 38 | 17500 |

predicted by the model that is inconsistent with the true label. the smaller the OneError, the higher the accuracy of the model in predicting the highest confidence label.

- Coverage (Cov): this metric measures the proportion of the number of labels that need to be considered in order to cover all positive labels. It reflects the efficiency of the model in the recommendation task. The smaller the coverage, the lesser the number of labels the model needs to consider in the recommendation task and the more efficient it is.

