# OpenReview forum: "Rare-Label-Oriented Discriminative Driven Feature Construction for Double Incomplete Multi-View Multi-Label Classification"
_ICLR.cc/2026/Conference — Submitted to ICLR 2026_

### Official Review · Reviewer_TtFv · 2025-10-15

**Soundness:** 3
**Presentation:** 2
**Contribution:** 3
**Rating:** 4
**Confidence:** 4

**Summary:**

This paper proposes a model named RLOD-net for the double incomplete multi-view multi-label classification (DiMvMLC) task, which involves both missing views and missing labels. The authors argue that existing methods over-rely on consistency information across views, neglecting view-specific information, which particularly impairs the recognition of rare labels. To address this, RLOD-net employs a dual-encoder framework to decouple shared and private features. It introduces a hierarchical contrastive learning loss to enhance feature separability. Furthermore, it uses a discriminative-driven mechanism to augment shared features and performs decision-level fusion to improve prediction performance on rare labels.

**Strengths:**

The paper is well-structured and highly readable. The experimental performance is notable, and the problem it addresses is practical.

**Weaknesses:**

Please see Questions.

**Questions:**

1)Regarding the fusion of shared and private information (Equation 12), which acts as a gating mechanism. Does this design introduce a risk where information from the private features P is lost or "gated off" if the shared representation C has values close to zero after the sigmoid function? Could you provide a more detailed intuition or theoretical justification for selecting this specific multiplicative gating mechanism?
2)The paper's core claim is being "rare-label-oriented." However, the shared representation C is learned from all labels, meaning its representational capacity could be dominated by high-frequency, common labels. After this potentially biased shared representation is integrated with private features, could it adversely affect the final classification performance on rare labels, even with the use of decision-level fusion at the end?
3)Section 2.5 and the title emphasize "rare-label," and the paper states it addresses the imbalance problem, yet this aspect receives limited discussion. How does the method define and embody the "rare-label" concept? This is explained in the appendix but could be emphasized in the main text. Furthermore, why is it claimed that private features can solve the imbalance problem when the algorithm does not seem to incorporate specific mechanisms to handle it?
4)The paper refers to "view-specific features" as "private." Is this a self-defined term for this work, or does it have a connection to concepts in other fields, such as federated learning?
5)Equation 5 is part of the strategy to enforce orthogonality between shared and private features. Given this, why does the first term of the negative sample loss not consist of negative pairs formed between private features p_i(v) and shared features c?
6)In Section 2.3.2, the authors construct a "label-level" contrastive loss. However, the granularity of both the label similarity matrix L and the feature similarity matrix F appear to remain at the instance level. Could you clarify the meaning of "label-level" in this context?
7)Is there a functional overlap between the loss in Equation 15 and Equation 10? Equation 10 does not specifically target only "shared labels." If there is an overlap, what is the justification for also computing and classifying private features?

---

### Official Review · Reviewer_z2ZV · 2025-10-30

**Soundness:** 2
**Presentation:** 3
**Contribution:** 2
**Rating:** 2
**Confidence:** 4

**Summary:**

This paper proposes a is a novel framework RLOD-net for double incomplete multi-view multi-label classification (DiMvMLC), addressing limitations of existing methods that mask view specificity and weaken rare label recognition. It decouples shared and private features via dual encoders, extracting shared/private semantics respectively. A hierarchical contrastive loss enhances feature separability by aligning cross-view consistency and label-guided semantic similarity. A discriminative-driven mechanism aggregates shared information, while rare-label-oriented decision fusion preserves view-specific rare labels. Experiments on five datasets show RLOD-net outperforms SOTA methods, validating robust DiMvMLC and rare label classification.

**Strengths:**

1. The paper tackles the challenging and practical task of DiMvMLC (Double Incomplete Multi-View Multi-Label Classification), where both views and labels are missing.
2. This paper identifies the weakness "the inability to recognize rare labels" in existing methods and proposes a solution specifically oriented toward solving it.

**Weaknesses:**

1.	The innovation and contribution of the paper are limited. The use of shared and private encoders to decouple consensus and specificity is an conventional and standard operation in multi-view learning. Additionally, The $\mathcal{L}_ {hc}$ is a simple weighted sum ($\alpha\mathcal{L}_ {ic}+\beta\mathcal{L}_ {lc}$) of two well-known concepts, lacking methodological depth.
2.	The proposed RLOD-net has a complex, multi-stage architecture with four main modules, including dual encoder, decoders, multiple classifiers (for $\mathcal{L}_ {dcr}$ and $\mathcal{L}_ {df}$), and a total loss function combining five distinct components ($\mathcal{L}_ {re}, \mathcal{L}_ {ic}, \mathcal{L}_ {lc}, \mathcal{L}_ {dcr}, \mathcal{L}_ {df}$). This increases computational overhead and implementation difficulty.
3.	The label-level contrastive loss ($\mathcal{L}_{lc}$) depends on a similarity matrix $L$ calculated from the incomplete ground-truth labels ($Y \odot G$). In scenarios with extremely sparse or noisy labels, this matrix $L$ could be an unreliable guide, potentially degrading the feature learning it is intended to improve.

**Questions:**

Please refer to the Weaknesses.

---

### Official Review · Reviewer_uoHk · 2025-10-31

**Soundness:** 3
**Presentation:** 3
**Contribution:** 2
**Rating:** 4
**Confidence:** 4

**Summary:**

This paper proposes a method to address the challenges of Double Incomplete Multi-View Multi-Label Classification characterized by missing views and sparse labels. The proposed RLOD-net introduces a dual feature extraction framework to decouple shared and private features, a hierarchical contrastive loss to enhance feature separability, and a rare-label-oriented decision-level fusion strategy to improve rare label classification. The authors claim that the model achieves superior performance across five benchmark datasets compared to eight state-of-the-art methods.

**Strengths:**

1. The paper’s emphasis on improving rare label classification is a relevant and underexplored challenge in multi-label learning. The decision-level fusion strategy for rare labels is an interesting idea.

2. The hierarchical contrastive loss function is a theoretically valid approach to improving feature separability by leveraging both instance-level and label-level contrastive learning. The inclusion of label correlation as a guiding signal is a reasonable extension.

3. The authors compare their method against eight state-of-the-art models and evaluate it on five common benchmark datasets. This provides a broad empirical basis for assessing the method.

4. The paper is well written and easy to read.

**Weaknesses:**

1. The rare-label-oriented strategy relies on a simple weighted fusion mechanism without introducing fundamentally new techniques for rare label representation or classification. The approach lacks sophistication and fails to address deeper challenges, such as label imbalance or the dependency between rare and common labels.

2. The hierarchical contrastive loss and dual feature extraction framework introduce significant computational overhead. The scalability of the method to high-dimensional or large-scale datasets (e.g., with thousands of views or labels) is not discussed or tested.

3. The model relies heavily on hyperparameters, which require careful tuning. The sensitivity analysis in Figure 2 shows that suboptimal parameters can lead to significant performance degradation. This dependence limits the method’s practicality in scenarios where hyperparameter tuning is infeasible.

4. The decision-level fusion strategy and hierarchical contrastive loss are not interpretable. There is no insight into why certain views or features are weighted more heavily or how rare labels are prioritized. This lack of interpretability diminishes the method’s applicability in critical domains like healthcare or autonomous systems.

**Questions:**

Please refer to the weaknesses above.

---

### Official Review · Reviewer_PyDq · 2025-10-31

**Soundness:** 3
**Presentation:** 2
**Contribution:** 2
**Rating:** 2
**Confidence:** 4

**Summary:**

This paper proposes RLOD-net, a framework designed for double incomplete multi-view multi-label classification (DiMvMLC), addressing challenges such as missing views, missing labels, and rare label classification. The framework introduces a dual feature extraction mechanism, hierarchical contrastive loss, and a rare-label-oriented decision-level fusion strategy, achieving state-of-the-art (SOTA) performance on five benchmark datasets. While the contributions are significant, the manuscript requires improvements in clarity, novelty justification, and experimental completeness to strengthen its impact.

**Strengths:**

1.The dual feature extraction framework enables effective separation of shared and private features, preserving both multi-view consistency and view-specific uniqueness.
2. The framework is benchmarked against eight strong baselines on five widely used datasets, demonstrating consistent SOTA performance across multiple metrics.

**Weaknesses:**

1.	How does the hierarchical contrastive loss improve feature learning compared to traditional contrastive learning approaches
2.	You are suggested to evaluating RLOD-net on diverse datasets such as multimodal datasets.
3.	The use of symbols is not clear enough. The use of uppercase, lowercase and bold symbols makes it easy for readers to get confused.
4.	The scalability of RLOD-net to large-scale datasets or high-dimensional feature spaces is not discussed.
5.	You should analyze the runtime and memory usage of RLOD-net compared to baselines on larger datasets.

**Questions:**

See Weaknesses

---

### Meta-Review · Area_Chair_6YLX · 2025-12-07

**Summary:**

- The paper presents limited novelty in both its methodological design and overall contribution. Several core modules (e.g., shared/private encoders and weighted fusion) closely mirror standard multi-view learning practices, and the “rare-label-oriented” aspect lacks a clearly differentiated mechanism beyond reweighting.

- The proposed approach suffers from high algorithmic complexity and has not been adequately evaluated on large-scale datasets. The multi-stage architecture and composite loss likely add substantial computational overhead, yet the paper does not report convincing runtime/memory profiling or scalability tests on larger, higher-dimensional, or more diverse datasets.

- The authors have not submitted a rebuttal to address these critical concerns. Consequently, key issues, e.g., design justification, robustness to sparse/noisy labels and hyperparameter sensitivity remain unanswered, reducing confidence in the method’s practical value.

Therefore, I decide to reject this paper.

**Reviewer Concerns:**

None. The authors have not submitted the rebuttal.

**Reviewer Scores:**

Scores: 2, 4, 2 ,4

All reviewers agreed that the paper falls below the acceptance bar for ICLR.

---

### Decision · Program_Chairs · 2026-01-26

Reject